# Targeting Inflammation with Galectin-3 and PIIINP Modulation Among ST-Segment Elevation Acute Coronary Syndrome Patients Underwent Delayed Percutaneous Coronary Intervention

**DOI:** 10.3390/biomedicines13020259

**Published:** 2025-01-21

**Authors:** Saskia Dyah Handari, Mohammad Saifur Rohman, Djanggan Sargowo, Dahliatul Qosimah, Bayu Lestari, Ricardo Adrian Nugraha

**Affiliations:** 1Medical Faculty, Ciputra University, Surabaya 60271, East Java, Indonesia; 2Department of Cardiology and Vascular Medicine, Faculty of Medicine, Brawijaya University—Saiful Anwar General Hospital, Malang 65145, East Java, Indonesia; ippoenk@ub.ac.id (M.S.R.); djanggan@ub.ac.id (D.S.); 3Department of Chemistry, Faculty of Sciences, Universitas Brawijaya, Malang 65145, East Java, Indonesia; aulanniam@ub.ac.id; 4Laboratory of Microbiology and Immunology, Faculty of Veterinary Medicine, Brawijaya University, Malang 65151, East Java, Indonesia; dahlia_qosimah@ub.ac.id; 5Division of Cardiovascular Sciences, University of Manchester, Manchester M13 9PL, UK; bayu.lestari@manchester.ac.uk; 6Department of Cardiology and Vascular Medicine, Faculty of Medicine, Universitas Airlangga—Dr. Soetomo General Hospital, Surabaya 60115, East Java, Indonesia; ricardo.adrian.nugraha-2019@fk.unair.ac.id

**Keywords:** colchicine, galectin-3, percutaneous coronary intervention, ST segment elevation myocardial infarction

## Abstract

**Background/Objectives**: ST-segment elevation acute coronary syndrome (STE-ACS) represents a significant global health challenge, with cardiac remodeling and fibrosis critically affecting recovery after percutaneous coronary intervention (PCI). Colchicine, known for its anti-inflammatory effects, may regulate key fibrotic markers such as Procollagen III N-terminal Propeptide (PIIINP) and Galectin-3. This study assesses colchicine’s effect on these biomarkers in STE-ACS patients undergoing delayed PCI. **Methods**: In this multicenter, randomized, double-blind trial, we examined colchicine’s impact on Galectin-3 and PIIINP in 164 STE-ACS patients undergoing early or delayed PCI. Patients received colchicine shortly after hospital admission. Biomarker changes were evaluated at 24 h and five days post-treatment using two-way ANOVA. **Results**: Clinical trials in the early PCI group revealed that Galectin-3 levels decreased significantly on day one (*p* < 0.01) and further on day five (*p* < 0.0001), indicating Primary PCI has benefits to inhibition of fibrosis beyond colchicine add-on treatment. But, in the delayed PCI group, Galectin-3 levels significantly increased on day one (*p* < 0.01), but the decrease observed by day five was not statistically significant. It is related that the benefits of colchicine treatment may exceed PCI implantation in preventing cardiac remodeling. In the delayed PCI group, PIIINP levels showed a significant reduction on day five (*p* < 0.0001). **Conclusions**: This Colchicine demonstrates novel efficacy in delayed PCI, with a significant increase in Galectin-3 and a sharp reduction in PIIINP, indicating its ability to control fibrosis. This positions colchicine as a breakthrough therapy for improving outcomes in STE-ACS patients with delayed intervention.

## 1. Introduction

Cardiovascular diseases (CVDs) remain a leading cause of mortality globally, contributing to approximately 33% of all deaths [1]. Among these, ST-segment elevation acute coronary syndrome (STE-ACS) poses a significant burden, with over 7 million deaths reported annually [2]. Despite advancements in treatment modalities, the development of heart failure following STE-ACS remains a critical concern, increasing mortality risk by three to fourfold [3,4].

Patients who undergo Percutaneous Coronary Intervention (PCI) tend to have a lower risk of complications compared to those receiving medication-only therapy [5]. In Indonesia, managing STE-ACS cases often involves delayed percutaneous coronary intervention (PCI), primarily due to the long distances between rural communities and specialized healthcare facilities, along with economic limitations. As a result, delayed PCI poses a significant challenge, especially for patients from lower socioeconomic backgrounds. This delay is associated with worse outcomes, including greater myocardial damage, increased fibrosis, and a higher risk of heart failure, highlighting the urgent need to address this issue in clinical practice [6].

The pathophysiology of post-STE-ACS complications involves complex processes of cardiac remodeling, particularly fibrosis, which significantly impacts patient outcomes [7]. This remodeling process, characterized by structural and functional changes in the myocardium, can lead to heart failure if it is left uncontrolled [8]. Recent evidence underscores the importance of the systemic immune-inflammation index (SII) and the systemic inflammatory response index (SIRI), which have proven useful for concurrently evaluating inflammatory and immune statuses, demonstrating predictive capabilities for adverse outcomes in patients experiencing STE-ACS [9]. The “smoker’s paradox” explaining a protective risk of smoking in the setting of STE-ACS relates to the importance of a younger age of smokers [10]. From a pathophysiologic perspective, smokers have increased platelet aggregation, increased fibrinogen, and decreased fibrinolytic activity, creating a state of hypercoagulability that predisposes to acute thrombosis, induces endothelial dysfunction and neutrophil activation, causes oxidant injury, increases fibrinogen levels, and causes platelet activation during inflammation process [10]. Recent research has highlighted the role of inflammatory mediators and fibrotic markers in this process, with Galectin-3 and Procollagen III N-terminal Propeptide (PIIINP) emerging as key biomarkers.

Colchicine, a well-established anti-inflammatory agent, has shown promise in modulating the inflammatory response following STE-ACS. Its potential to mitigate cardiac remodeling by influencing fibrotic processes presents an intriguing therapeutic approach [11,12]. Uncontrolled cardiac remodeling can culminate in heart failure, a condition where the heart is unable to pump blood effectively to meet the body’s demands [13]. Monitoring patients at high risk for this remodeling is crucial for guiding effective treatment strategies [14]. However, the optimal timing of colchicine administration, particularly in the context of delayed PCI, and its effects on specific biomarkers of fibrosis and inflammation remain unclear. Moreover, few studies have investigated the timing of colchicine administration in relation to PCI. Colchicine administered 6 to 24 h before PCI was found to reduce the rate of periprocedural myocardial injury [15]. The use of colchicine in PCI has been shown to reduce major adverse cardiovascular events (MACE) [16].

Cardiac fibrosis, a hallmark of adverse cardiac remodeling, results from the excessive deposition of extracellular matrix components, primarily collagen, in the myocardium. This process is associated with a decline in ventricular function and an increased risk of heart failure [17]. Elevated PIIINP levels in the blood indicate increased collagen synthesis, potentially a sign of cardiac remodeling [18]. Galectin-3 is a protein involved in diverse processes, including fibrosis, inflammation, and apoptosis [19]. Elevated galectin-3 levels in the blood indicate increased fibrosis and inflammation, which can also contribute to cardiac remodeling and heart failure [20].

Numerous studies have explored biomarkers like Galectin-3 and PIIINP, both linked to fibrotic processes and elevated in patients following STE-ACS [21]. While the role of these biomarkers in cardiac remodeling has been extensively studied, effective therapeutic strategies to modulate these pathways remain limited. This study aims to elucidate the cardioprotective effects of colchicine on Galectin-3 and PIIINP in STE-ACS patients undergoing either primary or delayed PCI. Our findings suggest that colchicine’s efficacy is more pronounced in delayed PCI settings, where it significantly modulates these biomarkers, offering a potential therapeutic advantage in scenarios where timely revascularization is not feasible. This research provides novel insights into the timing-dependent efficacy of colchicine in modulating post-STE-ACS cardiac remodeling, particularly in resource-limited settings where delayed intervention is more common.

## 2. Materials and Methods

### 2.1. Study Design and Participants

This multicenter, randomized, double-blind, placebo-controlled trial was conducted from September 2022 to February 2023 across five healthcare centers in East Java, Indonesia. We enrolled 164 patients with first-time ST-segment elevation acute coronary syndrome (STE-ACS). Inclusion criteria for this study were first-time STE-ACS patients between the ages of 30 and 65, presenting with chest pain lasting more than 30 min and ST-segment elevation ≥ 1 mm in at least two contiguous leads on a 12-lead electrocardiogram. Exclusion criteria included patients with previous myocardial infarction, severe renal or hepatic dysfunction, and those with contraindications to colchicine. Patients were stratified into early PCI (≤12 h) and delayed PCI (>12 h) groups based on their time to intervention. This study adhered to the Declaration of Helsinki and was approved by the Ethical Committee of Saiful Anwar Hospital (Approval No. 400/235/K.3/302/2020). The trial was duly registered with the International Study Registry ISRCTN (Registration Number: ISRCTN12958502, registered on 28 December 2023).

### 2.2. Randomization and Blinding

Participants were randomized using a computer-generated sequence to receive either colchicine or placebo, stratified by time to PCI (early: ≤12 h; delayed: >12 h). An independent research team managed the randomization process. Both medications were identically packaged and labeled by an external pharmacy to maintain the blinding of participants, healthcare providers, and outcome assessors.

### 2.3. Intervention

Patients received either oral colchicine (0.5 mg daily) or a matching placebo for 12 weeks, starting on the day of admission. A loading dose of 1 mg colchicine was administered 1–2 h pre-PCI, followed by a 0.5 mg maintenance dose starting one-hour post-PCI, which was continued daily for one month.

### 2.4. Sample Size Determination

The sample size was calculated based on the primary outcomes of changes in Galectin-3 and Procollagen III N-terminal Propeptide (PIIINP) levels. The sample size was calculated to detect a medium effect size (d = 0.5) with 80% power at a 0.05 alpha level, requiring 64 participants per group. The total sample size was increased to 164 participants to accommodate potential dropouts and non-adherence, distributed as 102 in the early PCI group (51 colchicine, 51 placebo) and 62 in the delayed intervention groups (35 colchicine, 27 placebo) (Figure 1).

### 2.5. Monitoring, Safety, and Adverse Events

A systematic record was maintained for all adverse events encountered during or subsequent to medication administration. Notable adverse effects deemed related to the study were promptly reported following standardized safety reporting protocols.

### 2.6. Data Collection and Outcome Measures

Baseline clinical and demographic characteristics were recorded at enrolment. Primary outcomes were changes in serum Galectin-3 and PIIINP levels at 24 h and 5 days post-PCI. Secondary outcomes included changes in left ventricular end-diastolic volume (LVEDV) and major adverse cardiac events (MACE).

### 2.7. Research Variables

Independent Variables: Administration of colchicine, including both loading and maintenance doses.

Dependent Variables: Changes in cardiac remodeling were assessed using echocardiographic measurements of left ventricular end-diastolic volume (LVEDV) and ejection fraction (EF).

Intermediate Variables: Levels of Galectin-3 and the collagen I to III ratio, assessed via Procollagen III N-terminal Propeptide (PIIINP).

### 2.8. Biomarker Quantification

Venous blood samples were collected at 24 h and 5 days post-PCI. Serum Galectin-3 and PIIINP levels were quantified using specific enzyme-linked immunosorbent assay (ELISA) kits (Human Galectin-3 ELISA Kit, BT Lab, Shanghai, China; Cat No: E1951Hu; Human Procollagen Type III N-Terminal Propeptide ELISA Kit, Cat No: abx152740).

### 2.9. PIIINP Measurement Protocol

The ELISA for PIIINP involved the preparation of standard, blank, and sample wells. These were incubated with designated reagents at 37 °C, followed by a series of washes. Detection reagents A and B were then applied. A colorimetric substrate solution was added to develop the color, which turned blue, and a stop solution was subsequently used to change the color to yellow, signaling readiness for optical density measurement at 450 nm.

### 2.10. Galectin-3 Measurement Protocol

For the ELISA for Galectin-3, all reagents and samples were first brought to room temperature. Samples were added to labeled 8-well strips and incubated with gentle shaking for 2.5 h at room temperature. Following this, the wells were washed, and a biotinylated Galectin-3 antibody was added. After further incubation and washing, a streptavidin solution was applied. The reaction was developed using a 3,3′,5,5′-Tetramethylbenzidine (TMB) substrate and stopped with a stop solution before measuring the absorbance at 450 nm.

### 2.11. Statistical Analysis

Data were analyzed using GraphPad Prism Software (version 8.40). Quantitative results were expressed as means ± SEM. Prior to conducting parametric tests, data about PIIINP and Galectin-3 levels were assessed for normality and variance homogeneity. The effects of colchicine compared to placebo at different time points were analyzed using two-way ANOVA with Bonferroni’s post hoc test for multiple comparisons. Significance was established at a *p*-value of <0.05. Additionally, adjustments for potential confounders, such as age, gender, and baseline severity of STE-ACS, were incorporated into the statistical models to enhance the validity and reliability of the findings.

## 3. Results

### 3.1. Baseline Characteristics

Of 196 STE-ACS patients assessed for eligibility, 164 patients met the inclusion criteria and were randomized (Figure 1). Baseline characteristics were well-balanced between groups (Table 1, Appendix A). The mean age was 56.93 ± 10.32 years, with a predominance of male participants (82.21%). No significant differences were observed in cardiovascular risk factors or infarct-related artery location between groups.

### 3.2. Effects of Colchicine on Procollagen Type III N-Terminal Propeptide (PIIINP) Levels

This study quantitatively analyzed serum concentrations of PIIINP to evaluate the antifibrotic effects of colchicine in STE-ACS patients following early PCI. At the initial 24-h mark post-treatment, PIIINP levels showed no statistically significant difference between the groups, with colchicine-treated patients presenting a mean PIIINP level of 1768.41 ± 132.08 pg/mL and the placebo group showing 1769.06 ± 135.61 pg/mL (*p* > 0.9999). However, a significant reduction in PIIINP levels was observed by day 5 in the colchicine group (1649.94 ± 162.25 pg/mL) compared to the placebo group (1828.54 ± 167.65 pg/mL, *p* < 0.0001). These results suggest a pronounced antifibrotic effect of colchicine, particularly evident in patients undergoing early PCI (Figure 2, Appendix A).

Further analysis evaluated colchicine’s efficacy in STE-ACS patients who received late revascularization. Similar to early PCI outcomes, colchicine significantly altered PIIINP levels at day 5 in the delayed PCI group. At 24 h, the colchicine-treated group’s PIIINP levels (461.26 ± 113.77 pg/mL) were similar to those in the placebo group (463.19 ± 89.29 pg/mL, *p* > 0.9999). However, by day 5, a significant difference was observed between the colchicine (451.48 ± 93.00 pg/mL) and placebo (494.91 ± 60.76 pg/mL) groups (*p* < 0.0001) (*p* < 0.0001) (Figure 3, Appendix A).

These findings suggest that the timing of colchicine administration may be crucial for its effectiveness in modulating fibrotic biomarkers and highlight the importance of early PCI in guiding therapeutic strategies in clinical settings. ns: not significant; ****: *p* < 0.0001 revealed a significant reduction in PIIINP levels, confirming colchicine’s antifibrotic effect.

### 3.3. Effects of Colchicine on Galectin-3 Levels

In the early PCI group, colchicine significantly reduced Galectin-3 levels at 24 h (461.94 ± 125.03 pg/mL vs. 531.61 ± 141.59 pg/mL, *p* = 0.0040) and 5 days (237.06 ± 63.18 pg/mL vs. 565.53 ± 103.41 pg/mL, *p* < 0.0001) compared to placebo (Figure 4, Appendix A). In the delayed PCI group, colchicine initially increased Galectin-3 levels at 24 h (1479.5 ± 104.25 pg/mL vs. 1382.45 ± 109.91 pg/mL, *p* = 0.0066) but showed a non-significant reduction by day 5 (1461.04 ± 114.94 pg/mL vs. 1470.75 ± 172.90 pg/mL, *p* > 0.9999) (Figure 5, Appendix A).

Our analysis revealed that colchicine administration significantly impacted Galectin-3 and PIIINP levels depending on the timing of PCI. In the delayed PCI group, Galectin-3 levels showed an initial increase at 24 h (*p* < 0.01), indicating heightened fibrotic and inflammatory activity. However, by day 5, colchicine significantly reduced PIIINP levels (*p* < 0.0001), suggesting its potential to modulate the later stages of fibrosis, even when intervention is delayed.

## 4. Discussion

The baseline characteristics of the 164 participants were uniformly distributed across all groups. This included a balanced demographic of males and females aged 30–65, all presenting with first-time STE-ACS. Key clinical parameters such as duration of chest pain, ST-segment elevation, and cardiac enzyme levels were recorded to ensure comparability at the outset. The demographic and clinical profiles confirm that the study groups were well-matched, supporting the integrity of the trial’s outcomes. Our study meticulously profiled the baseline characteristics of STE-ACS patients, capturing a demographic primarily in their late fifties, with a majority being male. Notably, smoking prevalence was markedly high, especially in the early PCI-placebo group, though not significantly different when compared with the late colchicine group [9]. The incidence of conventional cardiovascular risk factors such as hypertension, diabetes mellitus, and dyslipidaemia showed no statistically significant differences across groups, reinforcing the comparability of the cohorts. The involvement of the infarct-related artery, specifically the left anterior descending (LAD) artery, was variably noted across the groups, yet without significant differences, ensuring a balanced baseline for evaluating the effects of colchicine therapy [22].

Clinical outcomes following colchicine administration were assessed, focusing on changes in LVEDV and the incidence of MACE. The increase in LVEDV, an indicator of cardiac function deterioration, was observed in nearly half of the patients in the early placebo group, with similar proportions noted in the early colchicine group. The late treatment groups had lower LVEDV increase percentages, but these differences did not achieve statistical significance. MACE, a critical clinical endpoint, was notably absent in both early PCI groups. The late placebo group documented a singular MACE occurrence, whereas the late colchicine group reported none, underscoring colchicine’s safety profile. Overall, statistical analysis confirmed that colchicine’s effect on both LVEDV increase and MACE prevention was not significantly different from placebo, indicating its potential non-inferiority in managing post-infarction outcomes [22].

This investigation centers on patients with STE-ACS, defined as an ST elevation greater than 1 mm (0.1 mV) in two or more contiguous leads [23]. The process of cardiac remodeling post-STE-ACS is influenced by various risk factors, including age, obesity, diabetes, hypertension, dyslipidaemia, and chronic ischemia resulting from coronary artery disease, alongside lifestyle factors such as smoking [24]. Post-STE-ACS inflammation can persist due to ongoing parietal stress on the myocardial wall [25]. This stress activates major neurohormonal pathways, specifically the renin-angiotensin-aldosterone system (RAAS) and the sympathetic nervous system, exacerbating fibrosis and accelerating apoptotic cellular changes [26]. These physiological changes contribute to adverse ventricular remodeling, characterized by alterations in heart structure and function, significantly elevating the risk of heart failure and subsequent mortality [27].

Research by Cole et al. (2021) has shown that a low dose of colchicine (0.5 mg daily) administered over 22.6 months post-STE-ACS can significantly lower the risk of ischemic cardiac events, such as death, resuscitated cardiac arrest, and recurrent myocardial infarction, compared to a placebo. This finding supports the initiation of colchicine treatment within three days post-STE-ACS to mitigate the risk of subsequent cardiovascular events [28].

Our study reveals ground-breaking insights into the role of colchicine in modulating cardiac fibrosis and inflammation, especially in the context of delayed PCI for STE-ACS patients. While early PCI demonstrated reductions in Galectin-3 and PIIINP, signifying controlled fibrotic and inflammatory responses, the delayed PCI group presented a remarkable finding. Despite the heightened fibrotic activity marked by the significant increase in Galectin-3 within the first 24 h post-PCI, colchicine administration successfully reversed this trajectory by significantly lowering PIIINP by day five [29]. This indicates colchicine’s potential to intervene in the late-stage fibrotic process, offering protection even when revascularization is delayed—a critical advantage for regions where timely access to PCI is a challenge.

Elevated levels of PIIINP may signify not only the normal fibrogenesis associated with healing but also excessive collagen deposition and disorganized healing, which are indicative of an amplified pro-inflammatory response and extensive tissue damage within the myocardium. Such pathological remodeling can lead to contractile dysfunction and disturbances in cardiac electrophysiology, consequently reducing myocardial pump efficiency and increasing the risk of arrhythmias [30]. Furthermore, elevated PIIINP levels post-STE-ACS have been correlated with adverse structural changes in the heart, particularly poor left ventricular function within the first year following the infarction, irrespective of the initial size of the infarct [31]. This biomarker is critical in regulating cellular interactions and healing processes within the cardiac tissue [32].

Elevated galectin-3 levels, indicative of myocardial damage, can trigger an inflammatory cascade that promotes macrophage infiltration, fibroblast activation, and excessive collagen deposition, leading to interstitial [33,34]. Our findings suggest that colchicine effectively modulates these processes, highlighting its potential in managing cardiac remodeling and fibrosis post-STE-ACS. This study underscores the pivotal role of galectin-3 following cardiac injury, where macrophages release this lectin to facilitate the fibrotic process by promoting collagen deposition and scar tissue formation [29]. Galectin-3 further drives the differentiation of fibroblasts into myofibroblasts, central to scar formation and extracellular matrix remodeling, resulting in increased myocardial stiffness and thickness. Such structural changes contribute to diastolic dysfunction, often manifesting as a consequence of extensive myocardial scarring [35,36].

Furthermore, data from a preceding study indicated that colchicine substantially lowered NT-proBNP levels compared to placebo, suggesting a beneficial effect on cardiac hypertrophy and remodeling processes (unpublished data). NT-proBNP is a marker secreted by cardiomyocytes under stress conditions such as myocardial infarction, and it possesses vasodilatory, diuretic, and natriuretic properties that aid in reducing cardiac preload and wall stress [37].

These findings suggest that colchicine could be a powerful therapeutic tool, ultimately mitigating the adverse remodeling seen with delayed intervention. By targeting key fibrotic markers like Galectin-3 and PIIINP, colchicine offers a novel pathway to improve long-term outcomes in STE-ACS patients undergoing delayed PCI, making it a potential game-changer in the management of acute cardiac events where timely intervention is not feasible. Our findings further support the role of colchicine in modulating not only fibrotic biomarkers like Galectin-3 and PIIINP but also key anti-inflammatory markers such as IL-10, as previously reported in our earlier studies. The significant increase in IL-10 levels following colchicine administration in a hypoxia-induced inflammation model reinforces its potential to reduce pro-inflammatory cytokine activity and promote myocardial healing in STE-ACS patients [38].

The lack of significant differences in LVEDV changes and MACE incidence between colchicine and placebo groups in our study may be due to the relatively short follow-up period. Longer-term studies are needed to determine whether the observed biomarker changes translate into clinically meaningful outcomes. Several limitations of our study warrant consideration. First, we focused on two key biomarkers, and a more comprehensive panel that included additional inflammatory and fibrotic markers could provide a more complete picture of colchicine’s effects. Secondly, the authors do not have any data about the compliance of patients to participate in the study. Lastly, we had a very short follow-up period. Longer follow-up periods would be valuable in assessing the long-term impact of colchicine on cardiac remodeling and clinical outcomes.

## 5. Conclusions

Our study demonstrates that colchicine significantly modulates fibrotic and inflammatory markers in STE-ACS patients, especially those undergoing delayed PCI. Despite the initial increase in Galectin-3, indicating heightened fibrosis, colchicine reduced PIIINP by day five, suggesting its ability to mitigate late-stage fibrosis. These findings highlight colchicine’s potential as a therapeutic strategy to improve outcomes in delayed PCI, offering new hope for patients in regions with limited access to timely interventions.

## Figures and Tables

**Figure 1 biomedicines-13-00259-f001:**
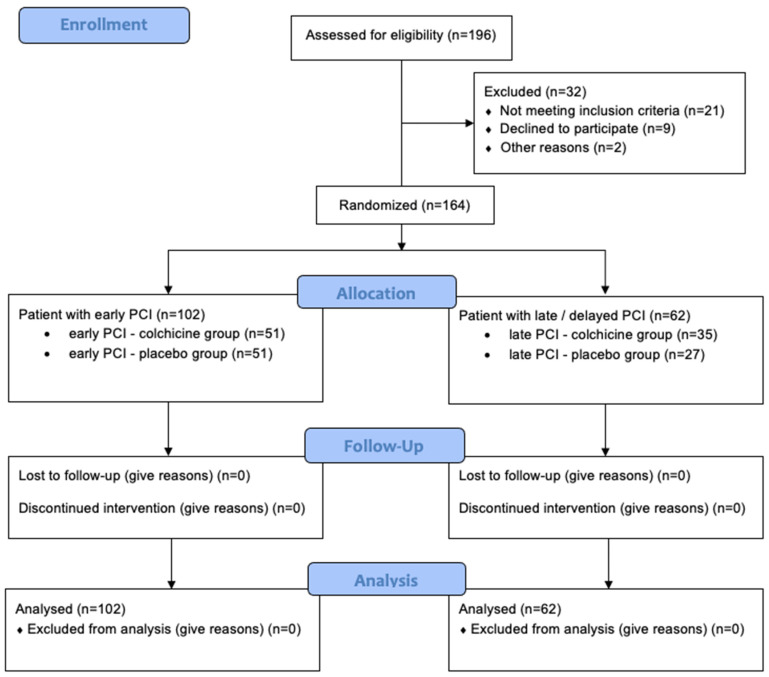
CONSORT 2010 Flow Diagram for our clinical trial study. The diagram illustrates the participant flow through each stage of the randomized trial, including enrolment, allocation, follow-up, and analysis. A total of 196 patients were assessed for eligibility.

**Figure 2 biomedicines-13-00259-f002:**
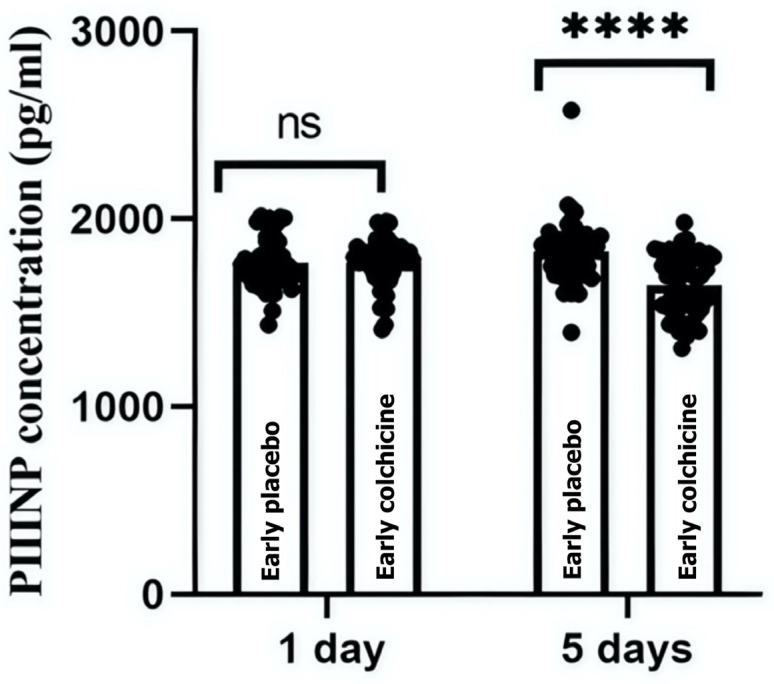
Impact of colchicine on PIIINP levels in STE-ACS patients undergoing Early PCI. This figure illustrates the effects of colchicine on serum PIIINP concentrations in patients receiving early PCI for STE-ACS. The graph compares PIIINP levels between the colchicine-treated and placebo groups at two time points: 24 h and 5 days post-MI. Initial measurements show no significant (ns) difference in PIIINP levels at 24 h; however, by day 5, colchicine treatment significantly reduces PIIINP levels, suggesting a potential therapeutic benefit in reducing cardiac fibrosis (**** *p* < 0.0001).

**Figure 3 biomedicines-13-00259-f003:**
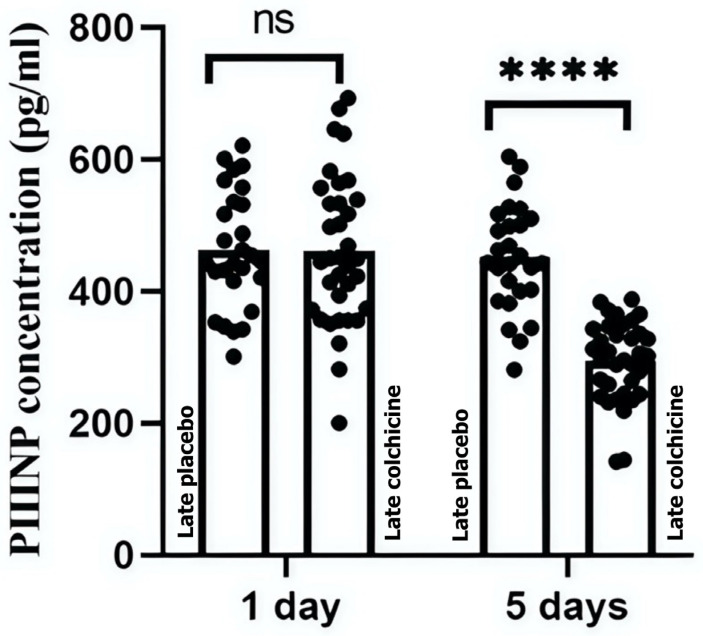
Colchicine’s effects on PIIINP in the delayed PCI treatment. This figure depicts the longitudinal effects of colchicine on PIIINP levels in patients with STE-ACS undergoing delayed PCI. Serum levels of PIIINP were measured at 24 h and 5 days post-intervention. The results indicate that colchicine does not significantly affect PIIINP levels at 24 h in this delayed intervention group; however, a significant difference is observed between the colchicine and placebo groups at day 5 (ns: not significant; ****: *p* < 0.0001).

**Figure 4 biomedicines-13-00259-f004:**
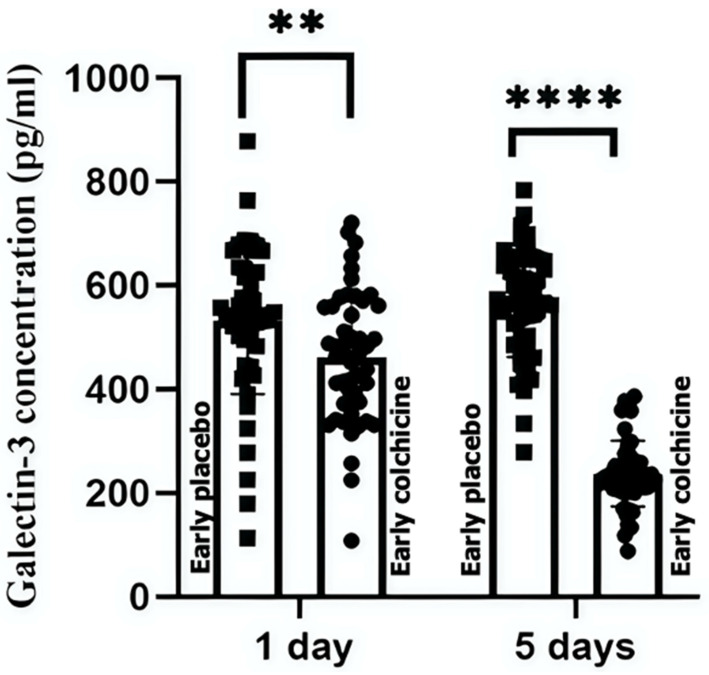
Impact of colchicine on serum Galectin-3 levels in STE-ACS undergoing early PCI patients. This figure illustrates the significant reduction in serum galectin-3 levels at 24 h and 5 days post-treatment in the colchicine-treated group, compared to the placebo group, highlighting its effectiveness in early PCI scenarios. These results underscore colchicine’s potential to influence cardiac fibrosis outcomes through galectin-3 modulation. (****: *p* < 0.0001, statistically significant; **: *p* < 0.01, statistically significant).

**Figure 5 biomedicines-13-00259-f005:**
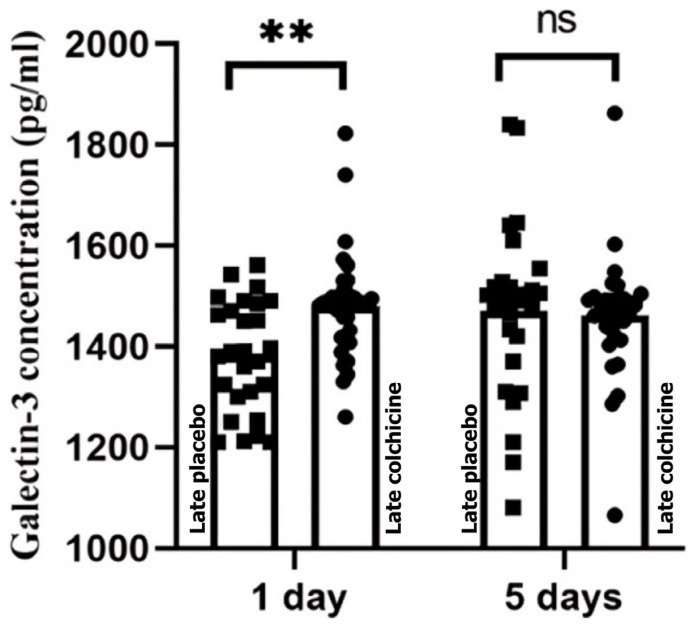
Effect of colchicine on Galectin-3 in patients with STE-ACS undergoing delayed PCI. This figure highlights the significant increase in serum galectin-3 levels at 24 h post-treatment in the colchicine group compared to the placebo group. By day 5, a non-significant reduction in galectin-3 levels was observed in the colchicine-treated patients compared to the placebo group. The data suggest a potential delayed anti-inflammatory effect of colchicine in modulating inflammatory pathways linked to cardiac remodeling in STE-ACS patients undergoing delayed PCI. (ns: not significant; **: *p* < 0.01, statistically significant).

**Table 1 biomedicines-13-00259-t001:** The baseline clinical characteristics of the study population.

	Early PCI-Placebo (*n* = 51)	Early PCI-Colchicine (*n* = 51)	*p*-Value	Late PCI-Placebo (*n* = 27)	Late PCI-Colchicine (*n* = 35)	*p*-Value
Age, *n* (%)	54.18 ± 10.83	59.00 ± 9.49	0.431	54.34 ± 11.91	60.20 ± 9.05	0.494
Sex (Male), *n* (%)	43 (84.31%)	39 (76.47%)	0.999	23 (85.19%)	29 (82.86%)	0.455
Smoker/Ex-smoker, *n* (%)	34 (66.67%)	32 (62.75%)	0.455	25 (92.59%)	30 (85.71%)	0.677
Hypertension, *n* (%)	33 (67.71%)	34 (66.67%)	0.786	13 (48.15%)	15 (42.86%)	0.999
Diabetes Melitus, *n* (%)	6 (11.76%)	12 (23.53%)	0.999	4 (14.81%)	5 (14.29%)	0.193
Dyslipidemia, *n* (%)	5 (9.80%)	3 (5.88%)	0.999	5 (18.52%)	11 (22%)	0.715
Infarct-related artery LAD, *n* (%)	30 (58.82%)	33 (64.71%)	0.999	11 (40.74%)	14 (40%)	0.684
Infarct-related artery non-LAD, *n* (%)	21 (41.18%)	18 (35.29%)	0.999	16 (59.26%)	21 (60%)	0.684

## Data Availability

All data underlying the results are available within the article and its Appendix A, further inquiries can be directed to the corresponding author.

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
