# Peer review of "Targeting Inflammation with Galectin-3 and PIIINP Modulation Among ST-Segment Elevation Acute Coronary Syndrome Patients Underwent Delayed Percutaneous Coronary Intervention"

_biomedicines, 2025, doi:10.3390/biomedicines13020259_

Round 1

Reviewer 1 Report

Comments and Suggestions for Authors

Although the article addresses the interesting topic of the effect of colchicine on the reduction of fibrosis in STE-ACS patients with delayed PCI, however, there are significant limitations to the publication of this article.

First, the groups were divided into those with early and delayed PCI. However, it was not stated how long the procedures were delayed on average in the second group, which can significantly affect the outcome and likelihood of fibrosis.

Secondly, there are major errors in the graphic parts of the article. The groups in the "Allocation" section do not match on Figure 1 and Table 1. Then, in Figure 2 and Figure 3, the "placebo" group and the colchicine "group" were labeled with the same labels. Furthermore, on Figures, "placebo" is misspelled as "plasebo".

Third, only LVEDV was used as an echocardiographic parameter of remodeling.

Fourth, the follow-up was too short.

On top of that, a recent study published in the NEJM showed that the use of colchicine in 7062 patients with acute myocardial infarction did not reduce the incidence of the composite primary outcome (death from cardiovascular causes, recurrent myocardial infarction, stroke, or unplanned ischemia-driven coronary revascularization).

Author Response

Although the article addresses the interesting topic of the effect of colchicine on the reduction of fibrosis in STE-ACS patients with delayed PCI, however, there are significant limitations to the publication of this article.

First, the groups were divided into those with early and delayed PCI. However, it was not stated how long the procedures were delayed on average in the second group, which can significantly affect the outcome and likelihood of fibrosis.

Authors' response: early PCI: ≤12 hours from the onset of chest pain, whilst delayed PCI: >12 hours from the onset of chest pain. We predicted a lack of benefit for delayed PCI in STEMI patients, but pathophysiological and retrospective observational studies suggest an advantage of colchicine for STEMI patients who experience delayed coronary revascularization.

Secondly, there are major errors in the graphic parts of the article. The groups in the "Allocation" section do not match on Figure 1 and Table 1. Then, in Figure 2 and Figure 3, the "placebo" group and the colchicine "group" were labeled with the same labels. Furthermore, on Figures, "placebo" is misspelled as "plasebo".

Authors' response: thank you for your suggestion. We apologise for the error in Figure 1, we have already corrected that issues. We have changed "plasebo" into "placebo" and labelled it newly in Figure 2, Figure 3, Figure 4, and Figure 5.

Third, only LVEDV was used as an echocardiographic parameter of remodeling.

Authors' response: We read a source stated that current definition of post ST‐segment elevation myocardial infarction (STEMI) left ventricular (LV) remodelling is purely structural (LV dilatation) and does not consider LV function (ejection fraction, EF). (Chimed S, van der Bijl P, Lustosa R, Fortuni F, Montero-Cabezas JM, Ajmone Marsan N, Gersh BJ, Delgado V, Bax JJ. Functional classification of left ventricular remodelling: prognostic relevance in myocardial infarction. ESC Heart Fail. 2022 Apr;9(2):912-924)

Fourth, the follow-up was too short.

Authors' response: yes, short follow-up duration is one of our study limitation. We have added in limitation subsection.

On top of that, a recent study published in the NEJM showed that the use of colchicine in 7062 patients with acute myocardial infarction did not reduce the incidence of the composite primary outcome (death from cardiovascular causes, recurrent myocardial infarction, stroke, or unplanned ischemia-driven coronary revascularization).

Authors' response: yes, i agree with the reviewer. Paper from Jolly et al (2024) showed that colchicine did not reduce the incidence of the composite primary outcome (death from cardiovascular causes, recurrent myocardial infarction, stroke, or unplanned ischemia-driven coronary revascularization). However, it reduces level of C-reactive protein as one of the inflammation biomarkers, similarly to Galectin-3 and PIIINP.

Reviewer 2 Report

Comments and Suggestions for Authors

We express our gratitude to the authors for submitting their manuscript to our journal. The article focuses on targeting inflammation through Galectin-3 and PIIINP modulation in patients with ST-segment elevation myocardial infarction (STEMI) who underwent delayed percutaneous coronary intervention. The topic is of significant interest, and the results presented are compelling and well-articulated.

We recommend several minor revisions to enhance the manuscript's clarity and impact.

First, the authors should consider creating a graphical abstract that concisely encapsulates the key findings and messages of the study.

Additionally, we advise condensing the abstract to fewer than 250 words while utilizing acronyms more extensively for improved readability. 

In the introduction, we suggest to discuss the role of inflammation assessment in STEMI patients, particularly concerning outcomes such as one-year mortality. We suggest referencing the work of Trimarchi et al. (PMID: 39458009, PMCID: PMC11508711, DOI: 10.3390/jcm13206059) on inflammatory indices, including the neutrophil-lymphocyte ratio and the advanced lung cancer inflammation index. 

Furthermore, a brief discussion on the controversial impact of smoking on mortality outcomes in STEMI patients, as well as its role in inflammation, would add depth to the manuscript. We recommend citing the article by Paradossi et al. (PMID: 38862370, DOI: 10.1016/j.carrev.2024.06.007) for further insight.

Author Response

We express our gratitude to the authors for submitting their manuscript to our journal. The article focuses on targeting inflammation through Galectin-3 and PIIINP modulation in patients with ST-segment elevation myocardial infarction (STEMI) who underwent delayed percutaneous coronary intervention. The topic is of significant interest, and the results presented are compelling and well-articulated.

Authors' response: thank you very much for your appreciation.

We recommend several minor revisions to enhance the manuscript's clarity and impact.

First, the authors should consider creating a graphical abstract that concisely encapsulates the key findings and messages of the study.

Authors' response: thank you for your suggestion. We will try to make a graphical abstract to help the readers easier to identify the information provided.

Additionally, we advise condensing the abstract to fewer than 250 words while utilizing acronyms more extensively for improved readability. 

Authors' response: thank you for your suggestion. We have reduced the abstract into 243 words.

In the introduction, we suggest to discuss the role of inflammation assessment in STEMI patients, particularly concerning outcomes such as one-year mortality. We suggest referencing the work of Trimarchi et al. (PMID: 39458009, PMCID: PMC11508711, DOI: 10.3390/jcm13206059) on inflammatory indices, including the neutrophil-lymphocyte ratio and the advanced lung cancer inflammation index. 

Authors' response: thank you for your suggestion. We have added paper from Trimarchi et al in the introduction section (reference number: 38).

Furthermore, a brief discussion on the controversial impact of smoking on mortality outcomes in STEMI patients, as well as its role in inflammation, would add depth to the manuscript. We recommend citing the article by Paradossi et al. (PMID: 38862370, DOI: 10.1016/j.carrev.2024.06.007) for further insight.

Authors' response: thank you for your suggestion. We have added paper from Paradossi et al in the introduction section (reference number: 39) for further insight.

Reviewer 3 Report

Comments and Suggestions for Authors

The topic of this peer-reviewed manuscript is definitely interesting and of potential value to “Biomedicines” readers.

Article „Targeting Inflammation with Galectin-3 and PIIINP Modulation among ST-segment Elevation Myocardial Infarction Patients Underwent Delayed Percutaneous Coronary Intervention” by Saskia Dyah Handari et al. is a multicenter, randomized, double-blind, placebo-controlled study investigating the cardioprotective effects of colchicine on galectin-3 and procollagen III N-terminal propeptide (PIIINP) in 164 patients with first ST-elevation acute myocardial infarction (AMI), who underwent either early or delayed percutaneous coronary intervention (PCI), focusing on its potential benefit in STEMI patients undergoing delayed PCI, where inflammation and fibrosis are more pronounced. Patients received colchicine shortly after hospitalization. Changes in biomarkers were examined 24 hours and five days after treatment. The study showed a novel efficacy of colchicine in delayed PCI with a significant increase in galectin-3 and a sharp decrease in PIIINP, indicating its ability to control fibrosis. This positions colchicine as a breakthrough therapy to improve outcomes in STEMI patients with delayed intervention.

1.       Original research articles should have a structured abstract of around 250 words.

2.       References number 1 and 2 do not refer to the assertion in the first two sentences of the main text? Revise the references in the main text.

3.       Typo error: periprocedural myocardial injury (PM) injury (line 82).

4.       In the Introduction the authors state: “Elevated galectin-3 levels in the blood indicate increased fibrosis and inflammation, which can also contribute to cardiac remodelling and heart failure”. However, it is worth noting that plasma galectin-3 was associated with heart failure and systolic dysfunction in patients six months after acute myocardial infarction (cite: PMID: 36672849 and 10.3390/biomedicines12122661).

5.       The Authors should add as limitation that they do not have any data about the compliance of patients to participate in the study.

6.       Abbreviations should be defined the first time they appear in each of three sections: the abstract; the main text; the first figure or table.

7.       Providing a more comprehensive explanation of the statistical methods used, including justifications for their selection, would greatly benefit the manuscript. Which tests did you use for normality and variance homogeneity?

8.       Figure 2 and Figure 3 are not self-explanatory; I cannot see the difference between the circles presenting the two groups of patients (colchicine-treated and placebo). You neglected to put a squares for one group.

9.       You wrote: “Contrary to early PCI outcomes, colchicine significantly altered PIIINP levels at both 24 hours and day 5 in these settings”. However, the following sentence “At 24 hours, the colchicine-treated group's PIIINP levels (461.26±113.77 pg/mL) were similar to those in the placebo group (463.19±89.29 pg/mL, p>0.9999)” and Figure 3 show the opposite.

10.    “Ns - not significant” and **** - p<0.0001 or ** - p<0.01 should be under the figures in a figure legend.

11.    Remove “highly” statistically significant as an explanation for ****; instead I suggest you write “**** - p<0.0001”.

12.    Is it possible for the data on LVEDV and the incidence of MACE to be presented in tables?

Author Response

  1. Original research articles should have a structured abstract of around 250 words.
    • Authors' response: We have corrected the abstract into a more structured abstract with  243 words
  2. References number 1 and 2 do not refer to the assertion in the first two sentences of the main text? Revise the references in the main text.
    • Authors' response: We apologised for the errors. We have corrected the references number 1 and 2.
  3. Typo error: periprocedural myocardial injury (PM) injury (line 82).
    • Authors' response: Thank you for your feedback. We apologised for the errors in line 82. 
  4. In the Introduction the authors state: “Elevated galectin-3 levels in the blood indicate increased fibrosis and inflammation, which can also contribute to cardiac remodelling and heart failure”. However, it is worth noting that plasma galectin-3 was associated with heart failure and systolic dysfunction in patients six months after acute myocardial infarction (cite: PMID: 36672849 and 10.3390/biomedicines12122661).
    • Authors' response: It is stated that Galectin-3 levels are increased in hypertrophy, fibrosis and inflammation. Also, the plasma concentration of gal-3 was significantly higher in patients who developed HF at 6 months (Dekleva, M.; Djuric, T.; Djordjevic, A.; Soldatovic, I.; Stankovic, A.; Suzic Lazic, J.; Zivkovic, M. Sex-Related Differences in Heart Failure Development in Patients After First Myocardial Infarction: The Role of Galectin-3. Biomedicines 2024, 12, 2661. )
  5. The Authors should add as limitation that they do not have any data about the compliance of patients to participate in the study.
    • Authors' response: thank you for your suggestion. We have added it in the limitation section. 
  6. Abbreviations should be defined the first time they appear in each of three sections: the abstract; the main text; the first figure or table.
    • Authors' response: thank you for your suggestion. We have showed the abbreviations in the first time they appear in the abstract, main text, and figures / tables.
  7. Providing a more comprehensive explanation of the statistical methods used, including justifications for their selection, would greatly benefit the manuscript. Which tests did you use for normality and variance homogeneity?
    • Authors' response: we used GraphPad Prism software (version 8.40). Normality test were used 1-sample Kolmogorov–Smirnov. All variables were normally distributed, then we decided to show the descriptive with mean + SEM. Following that, we performed two-way ANOVA with Bonferroni’s post hoc test for multiple comparisons.
  8. Figure 2 and Figure 3 are not self-explanatory; I cannot see the difference between the circles presenting the two groups of patients (colchicine-treated and placebo). You neglected to put a squares for one group.
    • Authors' response: thank you for your feedback. We have modified Figure 2-5 to make it easier to see the differences.
  9. You wrote: “Contrary to early PCI outcomes, colchicine significantly altered PIIINP levels at both 24 hours and day 5 in these settings”. However, the following sentence “At 24 hours, the colchicine-treated group's PIIINP levels (461.26±113.77 pg/mL) were similar to those in the placebo group (463.19±89.29 pg/mL, p>0.9999)” and Figure 3 show the opposite.
    • Authors' response: we apologise for the errors. The true statement is "Similar to early PCI outcomes, colchicine significantly altered PIIINP levels at day 5 in delayed PCI group"
  10. “Ns - not significant” and **** - p<0.0001 or ** - p<0.01 should be under the figures in a figure legend.
    • Authors' response: thank you for your suggestion. We have added it in the figure legends.
  11. Remove “highly” statistically significant as an explanation for ****; instead I suggest you write “**** - p<0.0001”.
    • Authors' response: thank you for your suggestion. We have removed the word "highly".
  12. Is it possible for the data on LVEDV and the incidence of MACE to be presented in tables?
    • Authors' response: unfortunately, the authors should say that there is lack of significant differences in LVEDV changes and MACE incidence between colchicine and placebo groups in our study due to the relatively short follow-up period, thus it is difficult to present it in tables.

Round 2

Reviewer 1 Report

Comments and Suggestions for Authors

Taking into account the defined limitations, the explanations offered, and the corrections made, this article may be published

Finally, I propose that STE-ACS replace the STEMI nomenclature

Author Response

Taking into account the defined limitations, the explanations offered, and the corrections made, this article may be published

Finally, I propose that STE-ACS replace the STEMI nomenclature

Authors' response: Thank you for your suggestion. We have modified STEMI into STE-ACS.

Reviewer 3 Report

Comments and Suggestions for Authors

1.       The authors have not included the suggested references in the main text.

2.       Indicate in the main text which tests you used for normality and variance homogeneity.

3.       What I had in mind for Figures 2 and 3 is to include squares (as you did for Figures 4 and 5) and to include the explanation in figure legends.

Author Response

  1. The authors have not included the suggested references in the main text.
    • Authors' response: thank you for your suggestion. We have included suggested references in the citation number 21 (Dekleva, M.; Djuric, T.; Djordjevic, A.; Soldatovic, I.; Stankovic, A.; Suzic Lazic, J.; Zivkovic, M. Sex-Related Differences in Heart Failure Development in Patients After First Myocardial Infarction: The Role of Galectin-3. Biomedicines 202412, 2661. https://doi.org/10.3390/biomedicines12122661)
  2. Indicate in the main text which tests you used for normality and variance homogeneity.
    • Authors' response: line 213-214, Prior to conducting parametric tests, data about PIIINP and Galectin-3 levels were assessed for normality and variance homogeneity
  3. What I had in mind for Figures 2 and 3 is to include squares (as you did for Figures 4 and 5) and to include the explanation in figure legends.
    • Authors' response: the dot markers / the squares come from the analysis. We decided not to include any figure legends, however we put the information inside the bar chart to help the readers easily identify which one is the early/late placebo and which one is the early/late colchicine group in day-1 and day-5